# The Many Ways to Deal with STING

**DOI:** 10.3390/ijms24109032

**Published:** 2023-05-20

**Authors:** Claire Coderch, Javier Arranz-Herrero, Estanislao Nistal-Villan, Beatriz de Pascual-Teresa, Sergio Rius-Rocabert

**Affiliations:** 1Departamento de Química y Bioquímica, Facultad de Farmacia, Universidad San Pablo-CEU, CEU Universities, Urbanización Montepríncipe, 28668 Boadilla del Monte, Spain; bpaster@ceu.es; 2Transplant Immunology Unit, National Center of Microbiology, Instituto de Salud Carlos III, 28220 Majadahonda, Spain; j.arranz3@usp.ceu.es; 3Departamento CC, Farmacéuticas y de la Salud, Facultad de Farmacia, Universidad San Pablo-CEU, CEU Universities, Urbanización Montepríncipe, 28668 Boadilla del Monte, Spain; estanislao.nistalvillan@ceu.es; 4Institute of Applied Molecular Medicine (IMMA), Department of Basic Medical Sciences, Facultad de Medicina, Universidad San Pablo-CEU, CEU Universities, Urbanización Montepríncipe, 28668 Boadilla del Monte, Spain

**Keywords:** STING, IFN, antiviral response

## Abstract

The stimulator of interferon genes (STING) is an adaptor protein involved in the activation of IFN-β and many other genes associated with the immune response activation in vertebrates. STING induction has gained attention from different angles such as the potential to trigger an early immune response against different signs of infection and cell damage, or to be used as an adjuvant in cancer immune treatments. Pharmacological control of aberrant STING activation can be used to mitigate the pathology of some autoimmune diseases. The STING structure has a well-defined ligand binding site that can harbor natural ligands such as specific purine cyclic di-nucleotides (CDN). In addition to a canonical stimulation by CDNs, other non-canonical stimuli have also been described, whose exact mechanism has not been well defined. Understanding the molecular insights underlying the activation of STING is important to realize the different angles that need to be considered when designing new STING-binding molecules as therapeutic drugs since STING acts as a versatile platform for immune modulators. This review analyzes the different determinants of STING regulation from the structural, molecular, and cell biology points of view.

## 1. Introduction

The stimulator of interferon genes (STING), also referred to as TMEM173, MPYS, ERIS, and MITA, is a homodimeric protein bound to the outer membrane of the endoplasmic reticulum (ER) of vertebrates. Different stimuli including DNA released by pathogens during infection, leak of nuclear or mitochondrial DNA, or the presence of cyclic di-nucleotides (CDNs) in the cytoplasm lead to the activation of STING, which results in the induction of a signaling cascade and the subsequent transcription of IFNB1 and a multitude of antiviral and proinflammatory genes [1,2].

The evolutionary origin of STING and its implications in antiviral mechanisms in different organisms have been reviewed recently by Cai and Ilmer [3]. Homologs of STING are present in many species [3]. The activation mechanisms of STING based on CDN recognition is well-conserved, even amongst some bacteria [4,5].

In mammals, STING is expressed in endothelial [6], epithelial [7], neuronal [8], and leukocytic cells, such as T cells [9], B cells [10], natural killer (NK) cells [11], macrophages, and dendritic cells (DC) [1]. STING has been described as one of the most important proteins involved in developing an antiviral response after recognizing pathogen-associated molecular patterns (PAMPs). It is involved in sensing the presence of foreign pathogens by reacting to CDNs produced endogenously by cyclic di-GMP-AMP (cGAMP) synthetase (cGAS) after binding to cytoplasmic double-stranded DNA (dsDNA) or CDNs produced by bacteria [12].

Cytosolic DNA receptors (CDRs) are pattern recognition receptors (PRRs) involved in the detection of DNA released by nuclear or mitochondrial damage or after pathogenic cell invasion. Many CDRs have been described previously. The cytosolic sensors that mediate the STING activation include cGAS; the four components of the absent in melanoma 2 (AIM2)-like receptors (ALRs), AIM2, interferon-inducible protein 16 (IFI16), interferon-inducible protein X (IFIX), DNA-dependent activator of IRFs (DAI), myeloid nuclear differentiation antigen (MNDA), helicases DHX9, DDX36, DDX41, and DExD/H box, nuclear enzymes DNA-PK, Ku70, and Ku80, RNA polymerase III, and meiotic recombination 11 homolog A (MRE11A) as reviewed by Paludan et al. [13]. Some other proteins have also been proposed as candidates for DNA sensing such as LRR binding FLII interacting protein 1 (LRRFIP1) [14].

The biological relevance of the correct activation of STING is highlighted by some STING-related interferonopathies [15]. In addition, several studies show that STING-deficient mice are more susceptible to viral infections by adenovirus and herpes simplex virus [16], as well as bacterial infections, because they cannot produce IFN-β and other cytokines in response to pathogen-derived DNA [17]. Infections by protozoan parasites can also be sensed through STING [18].

An increasing body of evidence suggests that STING also plays a role in the defense against RNA virus infections by directly detecting different viral-stimulated processes, or modulating the detection of viral RNA leading to the activation of the adaptor protein MAVS [19]. Evidence of the STING relevance against RNA viruses is highlighted by the presence of viral antagonists of the cGAS-STING pathway in different RNA viruses, such as the hepatitis C virus (HCV), yellow fever virus (YFV), or SARS-CoV-2 [16,20,21]. All this evidence shows that STING acts as a wide platform for the detection of pathogens and a promising target for triggering the antiviral response.

In addition, there are other fields where the development of STING-activating drugs has a great interest [22]. It has been demonstrated that the administration of STING agonist drugs in mice is a potent activator of the antitumoral response. Coadministration of STING ligands with other antitumor drugs such as anti-PD1/PDL1 or anti-CTL4 immunoglobulins leads to a robust immune response against tumors. For such a reason, there is a strong interest in developing STING agonists to be used in the clinic for the treatment of cancer patients [23,24].

This review focuses on the molecular aspects controlling STING regulation as it is regulated by numerous interactions with cellular and viral proteins. In addition, the many atomic resolution structures are sources of information to unveil the molecular basis controlling STING regulation. All this knowledge can be applied in developing STING ligands that can result in drugs able to modulate cellular processes involved in the induction of antiviral responses, control autoimmune diseases, or adjuvants for improving vaccination and cancer therapies.

## 2. cGAS-STING Canonical Signaling Pathway

Among all cellular DNA sensors, cGAS is the best characterized and considered essential for developing an innate immune response against cytosolic dsDNA [25,26]. This protein can detect pathogenic DNA or DNA released from the mitochondria or the nucleus after cell damage, even when it has been oxidized [25,27]. dsDNA fragments of more than 20 base pairs (bp) can be recognized by cGAS in a sequence-independent manner inducing its dimerization in a 2:2 DNA/cGAS complex [25,28]. Fragments of DNA smaller than 20 bp are recognized by cGAS but are unable to induce dimerization and thus do not induce its activation, while long chains of dsDNA induce the formation of ladder-like structures that result in a stronger activation of this sensor [25,29].

After the formation of dimers or higher structures, cGAS changes conformation enabling its catalytic site activation (Figure 1) [25,28]. Once activated, cGAS can use adenine triphosphate (ATP) and guanine triphosphate (GTP) as substrates to produce the secondary metabolite cyclic cGAMP [25,26,28]. cGAMP produced by cGAS is a CDN that contains two phosphodiester bonds, one between the 2′-OH of GMP and the 5′-phosphate of AMP and the other between the 3′-OH of AMP and the 5′-phosphate of GMP (2′3′-cGAMP). This ring-structured molecule acts as a second messenger, binding and inducing STING activation. It has also been described that cGAMP can pass to neighboring cells through gap junctions in a process dependent on connexin 43 allowing contacting cells to trigger STING activation [30].

When first described, STING was proposed to directly recognize bacteria-derived CDNs such as cdiGMP, cdiAMP, and 3′3′-cGAMP, which can act as PAMPs to activate the immune responses during infection [31].

In addition to interacting with STING in host cells, CDNs play a pivotal role as second messengers controlling physiological processes in bacteria [32,33]. Moreover, these molecules appear to modulate many behaviors facets at the bacteria community level, such as *quorum sensing*, the formation of aggregates, swarming motility, or the formation of bacterial biofilms [34,35]. There is a correlation between high intracellular levels of cdiGMP and biofilm formation and a sessile lifestyle. In contrast, low cdiGMP levels are associated with a motile or planktonic existence [36]. 3′3′-cGAMP also has been related to protection against phage infections in bacteria [37].

Inactive resting STING forms a homodimer that suffers a 180° twist and several conformational changes upon CDN-binding [38,39]. Conformational changes of STING after CDN activation promote the binding to the epidermal growth factor receptor (EGFR) in the ER, causing EGFR auto-phosphorylation at Y1068 and its consequent activation. Activated EGFR also phosphorylates STING at Y245 [40]. STING translocates from the ER to the Golgi apparatus by passing through the ER-Golgi intermediate compartment (ERGIC) [41]. The translocation process is driven by the STING-iRhom2-TRAPβ complex [42]. It is also known that K63-linked ubiquitination is required for STING trafficking, but it is not completely clear if this modification is produced on resting STING or after activation by CDNs [43]. 

Once in the Golgi, STING binds to sulfated glycosaminoglycans (sGAGs) and is palmitoylated in C88 and C91, with these two modifications being necessary to induce STING polymerization and the formation of oligomers [44,45]. Mukai et al. demonstrated by tritylation experiments that of the 23 human-encoded palmitoyl transferases (DHHCs), those responsible for the palmitoylation of STING are DHHC3, DHHC7, and DHHC15 [45]. This oligomerization is required for the signal transduction and activation of STING effectors [44,45]. It has been proposed that C88 and C91 palmitoylation induces the STING clustering into lipid rafts from the cytosolic side of the Golgi while sGAGs promote STING polymerization from the lumen [45]. After oligomerization, STING recruits and activates the downstream effector kinase TBK1 [28]. Activated TBK1 dimerizes and induces its autophosphorylation and phosphorylation of STING in two different residues, S358 and S366 [28,46,47,48]. As the first step, TBK1 phosphorylates S358, stabilizing the STING-TBK1 complex. The kinase domain of TBK1 cannot phosphorylate S366 since it is bound to this region of STING. To achieve S366 phosphorylation, TBK1 requires the previous formation of STING oligomers so that the kinase domain reaches S366 of the neighboring STING-TBK1 complexes [46].

The TBK1-STING phosphorylated complex recruits and phosphorylates Interferon Regulatory Factor 3 (IRF3) inducing its dimerization and translocation into the nucleus, which results in the expression of type I IFNs, a set of Interferon-Stimulated Genes (ISGs) and proinflammatory cytokines [49]. IRF3 is not the only transcription factor that STING can recruit. STING activation has been shown to induce the activation of IKKε, which in a redundant combination with TBK1 can induce the phosphorylation and translocation of NF-κB [50]. STING-dependent activation of NF-κB is responsible for the production of proinflammatory cytokines such as IL-6, IL-12, or TNF-α [28,51]. All these effects together lead to the development of an antiviral state and trigger innate and adaptive immune responses.

STING activation has been linked to other biological effects in addition to IFN and proinflammatory cytokines production. It is well known that STING activation triggers autophagy in a process that is independent of TBK1, IFN, and the Unc-51-Like Autophagy Activating Kinase (ULK) [28,52,53]. STING induction of autophagy is necessary for cytosolic DNA and virus clearance and the depletion of activated STING structures [30,52]. Once STING is activated, it initiates a translocation from ER to the Golgi apparatus as described before. During this process, STING passes through the ERGIC intermediate compartment in a coat complex protein II (COPII) and ADP-ribosylation factor (ARF) GTPase-dependent manner [52]. STING containing ERGIC induces light chain 3 (LC3) lipidation into LC3-II, a classic effector of autophagy, in a process dependent on ATG5 and WIPI2 [30,52,54]. Although autophagy has not been related to the induction of type I IFN expression, its implication in the IFN system seems to provide a downregulation of PRRs and receptors whose abundance is necessary to avoid tissue damage produced by excessive immune stimulation [55].

## 3. STING Non-Canonical Signaling Pathway

The activation of STING is possible even in the absence of cGAS, 2′3′-cGAMP, or other CDNs as inducers (Figure 2) [56,57]. Until now, different cGAS-independent mechanisms of STING activation have been described.

Interferon Gamma Inducible Protein 16 (IFI16) is a DNA sensor located in the cytosol and the nucleus and has been proven to be both a coactivator in the classic cGAS-STING pathway and a cGAS-independent STING inducer [58]. In the canonical pathway, IFI16 is necessary for the 2′3′-cGAMP production by stabilizing the cGAS–DNA interaction and for the STING activation of TBK1, at least in some cell types [58,59].

The proposed IFI16-dependent non-canonical activation of STING begins with polyADP-ribose polymerase-1 (PARP-1) and ataxia telangiectasia mutated (ATM) detection of damaged DNA in the nucleus. After this recognition, ATM phosphorylates p53, inducing the recruitment of IFI16. The complex p53-IFI16 translocases to the cytoplasm and is capable of interacting with TRAF6. Once activated, TRAF6 induces the formation of multiple K63-linked ubiquitin chains on STING. This ubiquitination leads to a non-canonical process of STING activation [58]. IFI16-dependent activation of STING takes place without STING phosphorylation in S366, TBK1 recruitment, or trafficking to ERGIC, leading to a predominant activation of NF-κB instead of IRF3 as compared to the canonical cGAS-STING pathway [58].

Another non-canonical activation of STING that has been proposed is through ER stress induction [57]. It has been suggested that ethanol can induce STING-TBK1-IRF3 activation in a not-very-well-characterized ER stress-dependent activation. This activation leads to IRF3-dependent Type I IFN induction and apoptosis through B-cell lymphoma 2 (Bcl2)-associated X protein (BAX) activation [57]. Many viruses, such as herpes simplex virus (HSV) or West Nile virus (WNV), amongst others [60], can induce ER stress during infection [61,62]. Thus, understanding how this ER stress-dependent activation of STING works can be important to understand better host defense against infections.

Calcium cell homeostasis seems to influence STING activation as well [57]. Stromal Interaction Molecule 1 (STIM1) is a Ca^2+^ sensor that is in the ER and interacts with inactivated STING. Due to this interaction, inactivated STIM1 acts as a natural inhibitor of STING oligomerization and trafficking [56,57]. STIM1 knock-out cells have basal levels of STING activation and IFN expression that are higher than those present in cells that present a normal STIM1 expression [56,57]. Under ER Ca^2+^ depletion conditions, STIM1 suffers a conformational change that enables the interaction with calcium release-activated calcium channel protein 1 (ORAI1) [63]. This way, STIM1 has been proposed separately to facilitate STING activation. As for the case of ER stress, many viruses have shown ER Ca^2+^ alterations during infection, making it another possibility for triggering innate immunity through these changes [64].

Different studies have linked STING activation to RNA virus infections [62,65,66,67]. A hint of the relevance of STING in the innate immune response to RNA virus infection is that many RNA viruses have developed mechanisms to inhibit STING pathway activation [68,69,70,71].

Mechanisms of STING activation by these viruses seem diverse, and some remain unclear. One of the most evident is the indirect activation of the classical cGAS-STING pathway by the detection of mitochondrial DNA (mDNA) released due to the increase in ROS and the activation of the inflammasome during dengue virus (DENV) infection [72,73]. Another example is the detection of damaged nuclear DNA as a result of SARS-CoV-2 spike protein expression associated with infection. Due to its main role as a fusion protein, the expression of the Spike under SARS-CoV-2 infection leads to the formation of multinucleated syncytial cells in which damaged nuclei produce micronuclei, which, in turn, are detected by cGAS [74,75]. Meanwhile, membrane fusion has been proven to induce cGAS-independent STING activation that leads to TBK1 and IRF3 activation and the type I IFN response [76,77]. Influenza A virus (IAV) blocks this signaling through a direct interaction between Hemagglutinin (HA) fusion peptide (FP) and STING, preventing TBK1 activation and STING phosphorylation [77]. The exact mechanism by which membrane fusion activates STING has not been addressed. Another role of STING under RNA viral infections seems to be blocking the translation of viral and host proteins [67]. This activation of STING is dependent on classic cytosolic RNA sensors and RLR but is MAVS-independent and does not result in IFN expression or autophagy induction [67]. Different studies have shown this link between STING and RLR, but the mechanism is still unknown [1,78].

## 4. STING Regulation and Inhibition Mechanisms

STING activation is regulated by several cGAS-STING signaling pathway regulating factors. Starting with cGAS, numerous factors such as nucleosomes, chromatin-binding proteins, or circular RNAs can block cGAS synthetase activity interfering with DNA binding [79,80,81]. The phosphorylation of cGAS by DNA protein kinase (DNA-PK) is another negative regulator of the pathway, which results in a reduction in the synthesis of 2′3′-cGAMP. A deficiency in DNA-PK has been shown to induce an increased inflammatory response signature in both mice and patients [82]. Other mechanisms are focused on ligand regulation; for instance, controlling the production of 2′3′-cGAMP, the active transport of 2′3′-cGAMP to bystander cells and the extracellular matrix, and the import and sense of extracellular 2′3′-cGAMP by an alternative splicing form of STING, as found in reviews by Liang et al. and Zhang et al. [83,84].

Regarding STING regulation processes (Table 1), protein activation by itself can trigger its degradation to avoid overactivation of the inflammatory pathway. As described above, the STING-dependent activation of autophagy is related to its degradation [30]. STING activation of TBK1 not only leads to the phosphorylation of IRF3 but also the autophagy receptor p62/SQSTM1. After being activated by TBK1, this protein detects K63-linkage ubiquitinated STING, which is involved with its activation, and induces its recruitment to p62 and Rab7-positive compartments. This process is essential for autophagy-dependent STING degradation in acidified endolysosomes [85,86].

However, direct degradation is not the only mechanism by which autophagy-related factors can regulate STING activation. It has been described that upon dsDNA sensing, autophagy-related gene 9a (ATG9a) acts as a regulator of STING interaction with TBK1 [87]. The absence of ATG9a results in a recruitment enhancement of TBK1 by STING and an aberrant induction of the production of Type I IFN and proinflammatory cytokines [87].

ULK1 is another autophagy-related factor that has been linked to the negative regulation of STING after activation. It has been shown that the production of 2′3′-cGAMP by cGAS and other CDNs inactivates AMPK, a natural repressor of ULK1 [88]. The consequent activation of ULK1 is, in turn, responsible for the phosphorylation of activated STING at S366, preventing TBK1-dependent activation of IRF3 and, thus, the Type I IFN induction [88].

Dephosphorylation is also a way to downregulate STING activity. PPM1A, a protein that belongs to the PP2C family of serine/threonine protein phosphatases, has been shown to reduce STING-dependent antiviral signal by the dephosphorylation of STING at S358 and TBK1 at S172. These modifications dampen STING oligomerization [48].

On the other hand, different factors have been associated with an upregulation of STING activity. Tripartite motif proteins (TRIMs) are a wide family of proteins, many of which are involved in antiviral activities and the regulation of antiviral signaling pathways, including cGAS-STING [89]. TRIM38 has been shown to SUMOylate both cGAS and STING (K338), preventing their degradation and increasing their activity and the production of IFN. This SUMOylation is removed at later stages of STING activation by SENP2 in a process that triggers its proteasomal degradation [90]. TRIM32 and TRIM56 have been related to the K63-linked polyubiquitination of STING at K20, K150, K224, K236, and K150, respectively. K63-linked ubiquitination has been demonstrated to increase STING interaction with TBK1 and its antiviral activity [91,92]. However, not all TRIMs have up-regulatory effects over STING. K48-linked polyubiquitination at STING K370 and K275 induced by the lung tissue-specific TRIM29 and the mouse-specific TRIM30α, respectively, has been shown to induce STING rapid degradation [93,94].

The Ring Finger Protein (RNF) family also has different members involved in the regulation of STING activity. Two members of the family, RNF5 and RNF90, induce K48-linked ubiquitination of STING in K150, which, as in the case of TRIM29, promotes its degradation [95,96]. Interestingly, RNF26 induces K11-linked ubiquitination in the same residue, K150, protecting it from the mentioned K48-linked ubiquitination and, thus, from degradation [97]. It has been shown that RNF115 also induces an up-regulatory effect over STING by triggering its K63-linked polyubiquitination at K20, K224, and K289 [98].

Different Ubiquitin Specific Peptidases (USPs) have been related to STING regulation as well. USP20 and USP44 increase protein stability by eliminating K48-linked ubiquitination, halting protein degradation [99,100,101]. In the USP20-dependent deubiquitination, USP18, an ISG involved in desensitization to IFN, acts as an intermediary by interacting with STING and promoting the recruitment of USP20 [99,102]. On the other hand, USP49, USP13, USP21, and USP35 have been shown to reduce type I IFN production of STING by removing some of the upregulating ubiquitinations, reducing at the same time TBK1 and IRF3 activation. USP49 has been described to act against K63-linked ubiquitination, USP13 against K27-linked ubiquitination, USP13 against both, and USP35 seems to remove K6-, K11-, K27-, K29-, or K63-linked polyubiquitin chains [103,104,105,106].

Other proteins involved in STING activity regulation are UBXN3B, MUL1, AMFR, CYLD, EIF3S5, OTUD5, and MYSM1. The two first catalyze the K63-linked ubiquitination of STING, increasing its interaction with TBK1 [43,107]. The UBXN3B interaction with STING seems to occur in combination with TRIM53 [107]. AMFR, which is located at the ER, induces the K27-linked ubiquitination of STING in a process that is dependent on INSIG1. Inhibition of this ubiquitination reduces TBK1 recruitment making cells more susceptible to viral infection [108]. In the case of CYLD, EIF3S5, and OTUD5, the upregulation of STING occurs via the removal of K48-linked ubiquitination, which protects STING from degradation [42,109,110]. EIF3S5 deubiquitination is produced after its recruitment by iRhom2 during STING trafficking, increasing its stability during this crucial step in the activation mechanism [42]. MYSM1, on the other hand, is produced after cytoplasmic DNA detection and acts as a negative regulator of STING by cleaving K63-linked ubiquitination at K150. This protein is decreased in systemic lupus erythematosus patients, resulting in an aberrant production of Type I IFNs and proinflammatory cytokines [111].

**Table 1 ijms-24-09032-t001:** Main post-translational modifications involved in the regulation of STING.

	Modification	Protein Involved	Locus (Human STING)	Outcome
**Positive regulation**	K63-linked ubiquitination	TRIM32 [91,92], TRIM56 [92], UBXN3B [107], MUL1 [43], RNF115 [98]	K20, K150, K224, K236, K289	Increase of STING-TBK1 interaction
	K27-linked ubiquitination	AMFR [108]	K137, K150, K224, K236	Increase of STING-TBK1 interaction
	K11-linked ubiquitination	RNF26 [97]	K150	Prevention of K48-linked ubiquitination
	K48-linked deubiquitination	USP20 [99,100,102], USP44 [101], CYLD [109], EIF3S5 [42] and OTUD5 [110]	K33, K236, K150, K347	Increase of STING stability
	Y245 phosphorylation	EGFR [40]	Y245	Enabling STING trafficking
	S358 phosphorylation	TBK1 [48]	S358	Enabling TBK1-STING complex formation
	S366 phosphorylation	TBK1 [46]	S366	Enabling STING interaction with IRF3
	Palmitoylation	DHHC3, DHHC7, DHHC15 [45]	C88, C91	Induction of STING oligomerization
	SUMOylation	TRIM38 [90]	K338	Prevention of STING degradation
**Negative regulation**	K48-linked ubiquitination	TRIM29 [93], TIM30α [94], RNF5 [95], RNF90 [96]	K370, K275, K150	Induction of STING degradation
	K63-linked deubiquitination	USP49 [103], USP21 [105], USP35 [106], MYSM1 [111]	K150	Decrease of STING interaction with TBK1
	K27- linked deubiquitination	USP13 [104], USP21 [105], USP35 [106]	Unknown	Decrease of STING interaction with TBK1
	K6-, K11- and K29-linked deubiquitination	USP35 [106]	Unknown	Decrease of STING interaction with TBK1
	S366 phosphorylation	ULK1 [88]	S366	Inhibition of STING-dependent IRF3 activation
	S358 dephosphorylation	PPM1A [48]	S358	Reduction of STING oligomerization
	deSUMOylation	SENP2 [90]	K338	Induction of STING degradation

Due to its evident role in dampening viral infections, viruses have developed numerous strategies to suppress cGAS/STING signaling. As it was mentioned before, the activation of STING can be triggered by the detection of viral DNA, self-DNA fragments derived from the nucleus (chromosomal DNA) or the mitochondria (mtDNA), or other different mechanisms, some of them not well defined. For this reason, viruses that evade cGAS-mediated innate immunity are not only strictly DNA viruses but also RNA viruses that can produce inhibitory proteins [19,112]. Viral strategies for inhibiting or dampening STING-dependent antiviral signals are diverse and target different functions of the pathway or trigger the direct degradation of the proteins involved. This review is focused on the inhibitors that target STING protein or its direct interaction with its upstream and downstream effectors. The viral inhibitors of cGAS have been recently reviewed by Hertzog and Rehwinkel [113].

Starting with the activation of the protein, vaccinia virus (VACV), amongst other viruses from the *Poxviridae* and other families, produces endonucleases, also referred to as poxins, which specifically degrade 2′3′cGAMP, inhibiting the activation of STING after the detection of viral DNA by cGAS [114,115]. 

Protein trafficking is also targeted by viral effectors for inhibiting the STING-dependent production of type I IFN. Human cytomegalovirus (HCMV) produces at least two different proteins that halt STING trafficking from the ER to the Golgi, namely UL94 and UL82. UL94 inhibits STING dimerization after activation, blocking protein trafficking, which is essential for the later activation of IRF3 or NF-kB [116]. UL82 has been shown as a disruptor of the formation of the STING-iRhom2-TRAPβ complexes needed for the translocation of the protein [117]. Similarly to the HCMV UL94 protein, the fusion peptide of the IAV HA protein blocks STING dimerization but only when the protein is activated in a non-canonical way by membrane fusion and not by 2′3′-cGAMP [77]. Herpes simplex virus (HSV) 1 and murine cytomegalovirus (MCMV) are also capable of inhibiting STING signaling at this step with their γ_1_34.5 protein [118]. 

Inhibition of protein oligomerization is another crucial step in the STING signaling pathway that viruses block to inhibit the production of IFN. Decreasing the K63-linked ubiquitination of STING is a common method of inhibiting oligomerization for different viruses. VP1-2 of HSV1, 3CL of SARS-CoV-2, the Polymerase of HBV, the TAX protein of human T lymphotropic virus type 1 (HTLV1), PLpro of human coronavirus NL63 (HCoV-NL63), and PLP2 of porcine epidemic diarrhea virus (PEDV) have been proven to block STING activation through this mechanism [21,119,120,121,122,123]. 

Another way to dampen the STING-mediated antiviral signaling is by interfering with the activation of the effectors downstream of this protein. HSV1 can modify the signaling cascade by binding TBK1 with its C-terminus and STING with its N-terminus [124]. Similarly, the ICP27 protein, also produced by HSV1, has been shown to bind both TBK1 and STING, thus blocking the activation of IRF3 [125]. Coronaviruses also have different examples of this kind of inhibition. The PLpro of SARS-CoV-1 interacts with the components of the complex STING-TRAF3-TBK1, halting the signal transduction [120], while ORF3a protein of SARS-CoV-2 binds to STING and has been shown to inhibit STING-dependent activation of NF-κB but does not interfere with the IRF3 activation [21]. Similarly to ORF3a, the Vpx protein from human immunodeficiency virus 2 and the simian immunodeficiency virus (HIV2 and SIV) binds STING in a domain that specifically interferes with NF-κB activation [126]. Kaposi’s sarcoma-associated herpesvirus (KSHV) has different proteins that interfere with the STING activation of its usual effectors. The protein vIRF1 has been shown to bind STING in a way that blocks its interaction with TBK1 [127]. ORF48 interacts with the host protein PPP6C inducing the dephosphorylation of STING and inhibiting the activation of the IRF3 signaling pathway, but not interfering with NK-κB activation [128]. Similarly, ORF33, also produced by KSHV, binds both STING and MAVS proteins, favoring the recruitment of PPM1G, which, in turn, dephosphorylates both proteins inhibiting the IFN production [129]. HCV has been proven to inhibit STING activation by interfering with its interaction with MAVS through the direct binding of its protein NS4B [20]. The NS2A protein produced by duck tembusu virus (DMTUV), a flavivirus, has been also reported to inhibit the activation of several proteins involved in the detection of PAMPS in ducks, including RIG-I, MDA5, MAVS, and STING. In the proposed mechanism for STING inhibition, DMTUV’s NS2A binds STING, acting as a competitor with both TBK1 binding and the STING-STING dimerization [130].

In addition to the inhibition of the different steps in STING activation, other strategies such as direct protein elimination have been discovered for different viruses. Several flaviviruses have been shown to induce the direct cleavage of STING, such as the NS2B3 protein of DENV, Japanese encephalitis virus (JEV), WNV, DMTUV, and Zika virus (ZIKV) [69,131,132].

Finally, an interesting way to reduce the amount of STING during infection is an increase in the expression of truncated isoforms of the protein, thus decreasing its activity in a mechanism that is still unknown. The production of these isoforms has been proven to be increased under the infection of different viruses such as the vesicular stomatitis virus (VSV) and the HSV1 [133].

## 5. Tridimensional Structure of STING and Conformational Changes upon Activation

Human STING is a homodimer in which each subunit is made up of 379 amino acids. The tertiary structure of STING includes a four-α helix amino-terminal transmembrane domain (aa 21–173, TM1 to TM4) in which several small loops are facing the interior of the ER, followed by a carboxy-terminal domain known as the Ligand-Binding Domain (LBD) or CDN-binding domain, located in the cytosol (aa 174–379) (Figure 3A). There have been many structures deposited in the Protein Data Bank (www.rcsb.org (last accessed on 15 March 2023)) of the LBD, but it was not until the work of Xiao-Chen Bai and Xuewu Zhang et al. in 2019 and 2022 that the full eukaryotic STING structure (human and chicken STING) was elucidated by cryo-EM (PDB codes 6NT5, 6NT7, and 7SII) [39,134]). To this date, there are 75 human STING crystal and Cryo-EM structures deposited in the PDB either in the apo form, bound to natural activators, or small organic drug candidate molecules (Appendix A).

The quaternary structure of STING is brought about by the dimerization of two monomers in a butterfly shape (Figure 3A). The dimerization involves the intertwining of the transmembrane α-helixes forming a sturdy base in which the two TM2 and TM4 interact with each other and are surrounded by the other two TM1 and TM3. Over this base rest the two “connector helixes” that span from P141 to G152 as found in PDB code 7SII, while the cytoplasmic ends interact adjacent to each other, forming the LBD (Figure 3A) [39]. The tertiary structure of the transmembrane N-terminal domain stabilizes the cytosolic LBD, and any mutation on the amino acids of the interface between domains alters the function of STING [39]. Upon ligand binding, what could appear to be a slight conformational change in the quaternary structure hides a major 180° rotation of the cytosolic domain over the transmembrane domain of each of the monomers as described above (Figure 3B). In the unbound conformation, LBDα1 of one of the monomers crosses LBDα1 of the other monomer; but when the CDN substrate binds, these two helixes are pushed apart and are oriented parallel to each other [39].

This behavior could not be properly seen in the many STING structures deposited in the PDB before 2019 due to the absence of the transmembrane domain. In addition to the 180° rotation, when bound to the CDN substrate, antiparallel sheets LBDβ4 and LBDβ5 are stabilized in each monomer and with the adjacent one, forming four antiparallel β strands, also known as the LBDβ4-LBDβ5 lid. Interestingly, when STING is not bound to the CDN, the cytosolic domain presents an open conformation in which sheets LBDβ4 and LBDβ5 have not been structurally resolved in any 3D structure, which leads to the proposition that they are disorganized in the apo form of STING (Figure 3E) [39,135]. Studies suggest that it may be mediated primarily by the symmetric interaction of R238 present in sheet LBDβ5 in both chains with the ligand. Upon ligand binding, the position of the side chain of R238 is stabilized and favors the formation of the β-sheet structures around the ligand, thus stabilizing a closed conformation [39]. Mutation studies carried out by Pu Gao et al. on human and murine STING propose that the transition from the open to the closed conformation of STING may not be the same in all species due to amino acid modifications in regions that are not implicated in the binding of the CDN [136].

Signal transduction brought about by STING activation has been proven to be through the oligomerization of STING dimers. This event has been observed in both prokaryotic and eukaryotic STING [5]. Moreover, there is a high homology in the CDN-binding domain amongst different species ranging from bacteria to humans [5]. By using the oligomerized chicken STING structure as a superimposition template, Morehouse et al. proposed that in ligand-bound bacterial STING, the structurally conserved LBD established protein–protein contacts between helixes α2 and α4 of the monomer of each homodimer. Furthermore, they subsequently proved that the oligomerization and activity of STING were altered by mutations in specific amino acids located in that region [5]. The oligomerization-dependent activity of STING has been found not only in bacterial STING but also in metazoans and chordates as it can be seen in the oligomerized structures of human and chicken STING (PDB codes 7SII and 6NT8) [39,134].

The closed and active conformation of human STING has also been found to oligomerize in at least a tetramer (dimer of dimers) by side-by-side packing. It has been suggested that the interaction between TM3 of the dimers in the tetramer can be implicated in this interaction, as well as the loop between LBDα2 and LBDα3, the latter agreeing with bacterial STING oligomerization. The conformational change suffered by STING in the bound conformation induces a change in the loop geometry that favors this side-by-side packing, whereas the unbound conformation would induce steric clashes [39]. 

The C-terminal ends that span from positions E339 to Q385 are located on the sides of each of the symmetrical chains of STING and are not present in any crystal structure due to their flexible nature. The recent cryo-EM structure of human TBK1 in complex with cGAMP bound to the full-length chicken STING (PDB code 6NT9) reveals that the C-terminal tail of STING adopts a β-strand-like conformation and inserts into a groove between the kinase domain of the second subunit and the scaffold and dimerization domain of the second subunit in the TBK1 dimer. Despite having used the full-length chicken STING, only amino acid positions 369 to 377 (which correspond to the same positions in human STING) have been solved. The amino acid sequence motif (D/E)XPXPLR(S/T)D is recognized by a TBK1-binding motif (TBM) and is conserved among different species [46]. Truncation of the 38 C-terminal residues does not affect cdiGMP binding [135], and mutations in the TBM lead to a complete loss of STING phosphorylation by TBK1 and a subsequent loss of downstream activity [46].

## 6. STING Polymorphisms

Human STING genetic variations (Figure 4) have been well documented. Four frequent haplotypes in humans are translated into different protein variants: R232H, R71H-G230A-R293Q (HAQ), G230A-R293Q (AQ), and R293Q. The major natural single nucleotide polymorphism (SNP) variant allele is translated in the protein polymorphism R232H. Histidine at STING position 232 occurs in ∼14% of the human population. Despite similar activity to recognize cGAS-produced 2′3′-cGAMP, H232 has a reduced response to bacterial and metazoan CDNs, decreasing the ability to activate IFN-β signaling when compared to the more frequent protein variant R232 [137]. Data from the 1000 Genome Project shows that the R71H-G230A-R293Q (HAQ) isoform is the protein variant resulting from the second most common human TMEM173 allele that occurs in 20.4% of the population. However, there are other alleles with a lower frequency, resulting in the protein variants G230A-R293Q (AQ) in 5.2% of the human population, and R293Q in 1.5% of the human population. Amongst them, the SNP resulting in R293Q dramatically decreased the STING stimulatory response to all bacterial ligands [138]. In the American population, the R232H allele is present in less than 50% of the population, while in Europeans, this allele is dominant. In Asians, the most common alleles result in protein variants R232/HAQ [139]. All these protein variants can bind 2´3´-cGAMP but have different activation responses to the different bacterial CDNs [140]. 

### 6.1. Gain-of-Function Mutations of STING

Aberrant STING activation has been associated with certain autoimmune diseases such as systemic lupus erythematosus, inflammatory vasculopathy, rheumatoid arthritis, Aicardi-Goutières, or other less frequent pulmonary and autoinflammatory diseases such as SAVI (STING-Associated Vasculopathy with onset in Infancy). SAVI diseases are caused by mutations of STING (V147L, N154S, V155M, and G166E), which have also been seen in familial lupus patients [141]. It has been proven that these mutations induce the STING activation pattern independently of the binding of 2′3′-cGAMP. In the absence of stimuli, STING translocates permanently from ER to the perinuclear microsomes, resulting in a constitutive Type I IFN expression [141].

From a structural point of view, all of the SAVI mutations are located at the dimerization interface: V147L is located in the connector helix, N154S is located in the small loop between the connector helix and the LBDα1, V155M is located at the beginning of the LBDα1, and G166E is at the middle of the LBDα1. Structural studies have proposed that the V155M mutation induces a tighter packing of both STING monomers. V155 is located at the LBDα1 and is part of a hydrophobic core known to favor STING dimerization [142]. Nadia Jeremiah et al. proposed that the only way to accommodate the side of methionine in this position is to favor closer hydrophobic interactions with the surrounding amino acids. Interestingly, V155M in one subunit would stabilize the position of M271 of the same subunit, thus increasing the sulfur–aromatic interaction established with the side chain of W161 of the other subunit [143]. To determine if SAVI-STING activation depended solely on the mutations, an R232A mutation that impairs 2′3′-cGAMP binding and 2′3′-cGAMP-dependent STING activation was tested. The results indicated that STING activation due to the SAVI mutations V147L, N154S, and V155M is independent of 2′3′-cGAMP binding as R232A had no effect on impairing STING gain-of-function [41].

SAVI mutations are not the only mutations that lead to STING-associated autoinflammatory symptoms. Isabelle Melki et al. described in 2016 a series of mutations present in three subjects that were related to their pathologies: C206Y, R281Q, and R284G. These amino acids are in the loop that separates LBDβ1 and LBDβ2, and the beginning of LBDα4, which is not involved in the dimerization interface. Additionally, the effect of mutations in the proximity of positions 206, 281, and 284 was analyzed, allowing the identification of the mutation at position D205 as a gain-of-function mutation [144]. More recently, Salla Keskitalo et al. reported the novel gain-of-function mutation G207E present in a large family that presented other SAVI mutations and SAVI-related pathologies [145]. Additional gain-of-function mutations S102P and F279L were identified in a 9-year-old boy that presented hyperinflammatory symptoms. The authors argued that the F279L mutation is located in the proximity of the SAVI-related N154 and V155 positions, which likely caused a similar effect to the V147L and N154S mutations. As for the S102P mutation, the authors argued that it could be involved in cell trafficking [146].

### 6.2. Loss-of-Function Mutations of STING

The dimerization interface of the cytosolic domain has proven to be critical in STING function as seen by the SAVI haplotypes. Interestingly, mutations have been described in this highly hydrophobic surface that results in a complete loss of function: V155R, W161A, and Y164A. The V155R introduces a large extremely hydrophilic amino acid that impairs normal protein–protein interaction, whereas W161A and Y164A lead to a loss of hydrophobic contacts that lead to the abnormal activity of the protein [142].

Jin, L. et al. determined that mutations R71H, G230A, and R293Q found in the active HAQ haplotype led to a defective IFN stimulation. Moreover, they pinpointed it on mutations R71H and R293Q, located in the cytosolic domain but very close to the membrane. The two original arginines are located very close to several cysteine residues (C88, C91, and C292), which would have their pKa modified by the strong positive charge of the guanidinium groups. The authors hypothesized that mutation of the arginines would lead to a loss of function of the cysteines. Additionally, C88S, C91S, and C292S reduced IFN production [147]. Interestingly, other authors have reported that mutations in the two membrane-proximal cysteine residues C88 and C91 suppress palmitoylation. These STING mutants cannot induce STING-dependent host defense genes because STING cannot be activated [148]. Indeed, Simone M. Haag et al. reported the inhibition of STING by two nitro-derivative compounds that specifically targeted palmitoylation-related C91 [149], and a similar outcome can be achieved with the inhibitor 2-bromopalmitate (2-BP) [45].

## 7. Structural Insight into Ligand-STING Interaction

The molecular insights into ligand binding can be used to improve drug design to achieve better STING induction or inhibition using artificial ligands. STING ligands can either be natural STING activators or small organic molecules designed as drug candidates. These drug candidates can be divided into two groups depending on their chemical structure: Modified CDN ligands and non-CDN ligands.

### 7.1. Cyclic Di-Nucleotides as Natural STING Activators

Throughout evolution, CDNs seem to have always been STING substrates [5]. Bacterial STING has been described to bind only cdiGMP [31]; as compared to vertebrate STING, which binds cdiGMP, cdiAMP, and 3′3′-cGAMP produced by bacteria [31,150,151], as well as 2′3′-cGAMP generated by cGAS as a response to the presence of DNA in the cytosol (Table 2) [31,152,153].

The ability to recognize the different kinds of CDNs constitutes an evolutionary improvement as it enables vertebrate cells to set a line of defense against intracellular bacteria [5]. Morehouse et al. determined the importance of several amino acids in the binding of different CDNs in bacterial and human STING. Interestingly, human STING (and chicken STING) does not establish specific recognition interactions with the nitrogenated bases of the CDNs as can be seen in PDB code 7SII (PDB code 6NT7 for chicken STING) [5,39,134]. In the binding mode of 2′3′-cGAMP to STING, the only clear interaction is the electrostatic contact established between the side chain of R238 and the phosphodiester backbone of 2′3′-cGAMP responsible for the stabilization of LBDβ4 and LBDβ5 into the dimerized four β sheet lid formed in the closed conformation of STING (Figure 5A) [39]. This type of interaction can be found in all structures of human STING in complex with 2′3′-cGAMP (PDB codes 4LOH, 4KSY, 5BQX, 6DNK, 6Y99, and 7SII) [38,134,154,155,156,157]. Apparently, the CDNs specificity was lost from bacterial STING, as in the latter, the arginine residue that would correspond to R238 instead of interacting with the phosphodiester backbone establishes very distinct specific recognition interactions with the nitrogenated bases of the CDNs [5]. This difference in recognition patterns can be seen in the human STING structure bound to 2′3′-cGAMP (i.e., PDB code 7SII) and in several cdiGMP-bound bacterial STING crystal structures such as *Sphingobacterium faecium* STING (PDB code 7UN9), *Myroides* sp. *ZB35* STING (PDB code 7EBL), or *Prevotella corporis* STING (PDB code 7EBD), amongst others (Figure 5B) [158,159]. Additionally, the equivalent in bacterial STING to LBDα3 is longer than in humans, which leaves less room for the cyclic 2´3´-cGAMP, thus making it impossible for bacterial STING to recognize it [5]. In this line, Kranzusch et al. were able to determine, by studying the binding of different CDNs to the *Nematostella vectensis* anemone, which presents more than 500 million years of evolutionary divergence from humans, that the ability to bind several CDNs resides in the ability of the STING dimer to adopt different closed conformations able to adapt to the size difference depending of each CDN [153]. The plasticity of STING when binding to CDNs is also visible in the ability to bind 2′2′-cGAMP as found in PDB code 4LOI [154], and 3′3′-cGAMP as found in PDB code 6YDZ [157]. Strikingly, and in contrast to what appeared to be a straightforward activation-linked conformational change from the inactive open conformation to the active closed conformation, non-mutated human STING binds to cdiGMP in an open conformation as can be seen in PDB codes 4EF4, 4EMT, and 4F9G [135,142,160].

### 7.2. Small Organic Molecules as Drug Candidates

The therapeutic targeting of the cGAS-STING pathway is a current hot topic in cancer immunotherapy and the treatment of autoimmune diseases [28,161]. The evidence of the implication of the STING-cGAS pathway in the innate immune response has led to the exploration of the possible use of STING activators as adjuvants in vaccines [162,163,164,165]. However, this strategy has to be approached with caution as, normally, STING activators present poor pharmacokinetic properties and can induce systemic toxicity [164]. One of the main reasons for the poor pharmacokinetic profile of CDNs is the low cell permeability and their enzymatic degradation [166]. For this reason, there are two approaches to the development of STING activators: The design and synthesis of modified CDNs and the design and synthesis of non-nucleotide STING-activating ligands. To date, there are more than 10 activating ligand candidates in clinical trials, half of which are modified CDNs and the other half are non-CDNs [167].

**Modified cyclic di-nucleotides.** (Figure 6) The degradation of CDNs is carried out by the ubiquitous ecto-nucleotide pyrophosphatase/phosphodiesterase (ENPP1) responsible for the hydrolysis of 2′3′-cGAMP [166], but unable to cleave bacterial 3′3′-cGAMP. Thus, the first attempt at the obtention of synthetic STING activators was the obtention of modified CDNs. These modifications were either in the phosphodiester bonds giving rise to the phosphothionate, boranophosphates and carbamide, and thiocarbamide CDN families; modification of the nitrogenated bases (as can be found for ligands X5J, X5D, X4M, OK6, KWF, KWO, and KXD in PDB codes 7KVX, 7KVZ, 7KW1, and 8A2H, 8A2I, 8A2J and 8A2K, respectively) [168,169,170,171]; or, modifications in the ribose. Modifications in the ribose could be by either a change in the sugar as in the case of 3´3´-cdiaraAMP (PDB code 7OB3); by the addition of different types of substituents to the 2′ and 3′ positions (ligands KT8, GGF, OOE, M8T, PWB, PWB, 9UR, and 9UH found in PDB codes 6S27, 6YDB, 6YEA, 6Z0Z, 6YWA, 6YWB, 7SHP, and 7SH0, respectively); by restriction of the conformational freedom of the nucleotides (Locked Nucleic Acid approach) (as can be found for ligand V5V in PDB codes 6XF3 and 6XF4) [172]; or by the addition of a vinyl moiety at the C5′ position of one of the sugars (ligand 98F found in PDB code 7Q85). It is important to state that the research papers related to PDB codes 6YWA, 6YWB, 7SHP, 7SHO, and 7Q85 have not yet been released [157,168]. Moreover, modified CDNs that combine modifications in the phosphodiester bond, the ribose, and/or the nitrogenated base have been described in the literature (ligands ZEV (also known as MK-1454), X5J, X4M, and R4T found in PDB codes 7MHC, 7KVX, 7KW1, and 7A90, respectively; for the latter, the research paper has not yet been released) [170,173].

The best example of this type of approach is clinical candidate ADU-S100 (ligand GJF found in PDB code 8B2J (primary citation not yet published)), which entered a clinical trial for Patients with Advanced/Metastatic Solid Tumors or Lymphomas but was terminated by Novartis due to the lack of enough activity or efficacy [174,175]. Most recently, novel macrocyclic di-nucleotide derivatives tethered by the amine at position 6 of the purine rings have been described and crystalized (ligand V5V found in PDB codes 6XF3 and 6XF4) [172].

**Non-cyclic di-nucleotides.** (Figure 7) Orally bioavailable small-molecule compounds able to trigger type I IFN production have been under study since the 1970s. However, in many cases, despite being able to induce IFN expression in mice, compounds failed to produce such results in humans. One of the first molecules to be studied was CMA (10-carboxymethyl-9-acridanone), which strongly induces IFN-β in primary mouse macrophages and failed to elicit detectable antiviral responses in vivo [176]. Cavlar et al. proved that CMA induced the IFN response by binding to murine STING and that the amino acid composition of the dimerized four β sheet lid of the LBD (LBDβ4 and LBDβ5) was responsible for the activity in murine STING but not in human STING [176]. Flavone acetic acid (FAA) was found to be active in a screening in mouse solid tumor models and dimethyloxoxanthenyl acetic acid (DMXAA), and a xanthone-derived agonist obtained by active synthesis reached clinical trials [177]. They both were active on murine STING but showed no binding to human STING [178,179]. Attempts to generate DMXAA derivatives able to bind human STING have been made [180,181]. Pu Gao et al. proved that albeit that DMXAA established the same interactions with the binding site in both human and murine STING, S162A-, G230I-, and Q266-mutated human STING was able to effectively bind DMXAA [136]. This pointed in the direction that not only are high-affinity interactions needed to achieve STING activation, but also the ability of STING to adopt and stabilize the closed conformation. In the same line, α-Mangostin, a natural molecule that also presents a xanthone skeleton and is known to present antitumor, antiviral, and immunomodulatory effects, shows higher affinity to human STING than to murine STING, but weaker potency for inducing type I IFN compared to 2′3′-cGAMP in reporter assays [182,183].

Given the species specificity of the previously described STING activators, high-throughput virtual screenings of chemolibraries such as the ZINC database [184] and high-throughput in vitro screens on human cell lines were carried out to find new candidates. This latter technique enabled the description by Sali et al. of compound G10, the first activator of human STING able to trigger IRF3/IFN-associated transcription [185,186]. Using the same approach, Ramanjulu et al. also rendered the obtention of amido benzimidazole compounds able to displace 2′3′c-GAMP-binding and induce the IFN response. Crystallographic studies prove that two molecules of ligand HGJ (also known as Compound 1) bound a single STING dimer (PDB code 6DXG), so they proceeded to synthesize ligand HG4 (also known as Compound 2), which tethered two HGJ molecules, thus describing that HG4 was more potent than HGJ in inducing the STING-mediated IFN response (PDB code 6DXL). In contrast to the two previous compounds, Compound 3, which was not initially crystallized bound to STING showed that it was able to trigger an immune response and achieve complete tumor regression. Compound 3 was renamed HB3089 and underwent further studies as a dose-dependent STING agonist and is currently in pre-clinical development. Very recently, the cryo-EM structure of Compound 3 bound to human STING was obtained (ligand WJ6 in PDB code 8GT6) [187]. Strikingly, and in contrast to what was a common assumption, structural studies showed that ligands HGJ and HG4 stabilized an open conformation of STING, the lid was not resolved, and the LBDβ4 and LBDβ5 did not form in either of the monomers. Additional hydrogen-deuterium (HD)-exchange mass spectrometry showed that the lid behaved as in apo STING rather than as in 2′3′-cGAMP-bound STING, which suggests that lid-closing and STING-closed conformation is not needed for STING activation [188]. Ligand WJ6 binds similarly to HGJ and HG4, destabilizing the LBDβ4 and LBDβ5 that was not visible in the 3D structure. However, despite not stabilizing the closure of the CDN-binding site, the binding of WJ6 induces a conformational change in STING dimers that is translated into the approximation of both LBDα1 helixes. This same conformational change can be found in the constitutively active human STING bearing the V147L mutation present in the SAVI haplotype (PDB code 8GSZ) [187]. The same year that Compounds 1, 2, and 3 were first described, a series of tetrahydroisoquinoline derivatives obtained by hit-to-lead optimization after the hit was identified by Automated Ligand Identification System (ALIS) were reported as mild STING inhibitors, as they bound the open conformation of STING and blocked not only 2′3′-cGAMP binding but also the IFN response (Compound 18 (K5S) and Compound 1 (K5P) found in PDB codes 6MXE and 6MX3, respectively) [189]. This is indicative of the complexity of the conformation-related activation of human STING by non-CDNs in a similar manner to that of CDNs. However, the ability of STING activators to engage the LBDβ4 and LBDβ5 lid depends on the ability to interact with R232 and/or R238 as occurs with the carboxylic acid of compound 11 described by Cherney, E.C et al. (ligand B7L found in PDB code 7SSM) [190]. Most recently, following the open path set by the linked amidobenzimidazoles [188], the novel SHR1032 STING agonist has been described to activate anti-tumor immunity in several haplotypes. Chunying Song et al. described a series of fused tricyclic compounds of which Compound 2 (ligand GD2 found in PDB code 7T9U) and SHR1032 (ligand GC0 found in PDB code 7T9U) have been crystalized with human STING. Two molecules of ligand GD2 bound to the CDN-binding site stabilizing an open conformation of STING. Interestingly, the pyrazole moiety establishes a hydrogen bond with the side chain of S162, which is buried deep in the CDN-binding site, while the carboxamide moiety of each GD2 unit establishes two hydrogen bonds with the carbonyl and NH of the backbone of S241, thus impairing the formation of the LBDβ4-LBDβ5 lid. The structure of ligand GC0 is the result of combining ligand GD2 with an amidobenzimidazole derivative through a flexible propyl linker. The binding of GD0 to human STING is comparable to the previously tethered amidobenzimidazoles and to that of GD2, as it stabilizes an open conformation of STING and establishes the same key hydrogen bonds as GD2, thus destabilizing the LBDβ4-LBDβ5 lid [191].

The screening of a library of approximately 2.4 million compounds identified a set of molecules that included benzothiophene oxobutanoic acid, herein MSA-2 (ligand QAD found in PDB code 6UKM), which led to the synthesis of several tethered analogs (compounds QAV, QB1, QB7, QBA, QBD, QBG, and QBJ found in PDB codes 6UKU, 6UKV, 6UKW, 6UKX, 6UKY, 6UKZ, and 6UL0, respectively). MSA-2 was found to be highly permeable and thus orally active and able to inhibit 2′3′-cGAMP binding. Crystallographic studies (PDB code 6UKM) showed a similar behavior to that of ligand HGJ, as two MSA-2 molecules bound to a single human STING homodimer, but in this case, MSA-2 and the set of analogs stabilized the closed conformation of STING with the formation of the dimerized four β sheet lid. The binding was stabilized by the π-π stacking of the benzothiophene moieties of the two MSA-2 molecules, and the staking with Y167. Additionally, electrostatic interactions established with S162, A238, and T263 stabilize the binding. Very similar behavior was found for tethered analogs found in PDB codes 6UKU, 6UKV, 6UKW, 6UKX, 6UKY, 6UKZ, and 6UL0 [192].

Very recently, an orally available MSA-2 analog, compound BSP16, containing a Se atom instead of S, was described by Xi Feng et al. as a potent STING agonist with a good pharmacokinetic profile and durable antitumor activity. The crystal structure deposited by the complex of human STING with compound BSP16 (ligand A9X in PDB code 7 × 9Q) shows the stabilization of STING in the closed conformation enclosing two BSP16 molecules interacting with each other through π-π stacking interactions. Stabilization of the binding mode by the electrostatic interactions established with the side chain of R238 and van der Waals contacts with the side chain of I235 enhances the interaction [193]. Chin et al. described compound SR-717, a STING inducer with antitumor activity in both mouse and human STING (ligand V67 PDB codes 6XNN and 6XNP, respectively). As other non-CDN inhibitors do, SR-717 binds to STING using two molecules. The binding mode occurs by face-to-face π-π stacking interactions established by their difluorophenyl moieties while engaging an electrostatic interaction with the side chain of R238 and an additional π-π stacking established between the pyridazine moiety and Y167 [194].

Quite recently, a new and undescribed ligand-binding site located in the transmembrane domain has been determined by cryo-EM (PDB code 7SII). Compound 53 (C53) (ligand 9IM found in PDB code 7SII) was found to be facing the lumen of the ER in a small pocket made up of TM2 and TM4 and the TM3-TM4 loop of one of the monomers (the second not being visible in the cryo-EM structure). Interestingly, in contrast to the lone binding of 2′3′-cGAMP, the binding of C53 induces the oligomerization of STING due to the induced fit brought about by the binding of C53 to the transmembrane domain. The binding of C53 pushes TM2 outward, thus apparently favoring the interaction between monomers [134].

However, not all the described ligands that can bind STING have been crystallized, but the existing crystal structures of human STING have allowed their identification despite their activity being tested in mouse models. A High-Throughput Virtual Screening (HTVS) of approximately 500,000 compounds extracted from the ZINC database was carried out by Ze Hong et al. on the CDN-binding site of the opened conformation of human STING found in PDB code 4EF5. The authors described that they obtained the binding of similar structures, of which they chose the eleven highest-ranking compounds for in vitro evaluation. Of these compounds, the compound identified as SN-011 showed inhibition activity against both WT and the SAVI haplotype in mouse models [195]. Jung Long et al. reported in 2022 a series of fusidic acid (FA) derivatives as STING inhibitors for the prevention of sepsis. The “cytokine storm” produced by inflammation has been determined to be the most critical aspect of sepsis. The authors reported a series of FA derivatives that presented better anti-inflammatory properties than FA. The most active, Compound 30, was docked in the CDN-binding site of the STING PDB structure 4KSY, whereas the anti-inflammatory effect was tested in mouse models. Despite not being tested in human STING, Compound 30 would be a promising STING inhibitor [196].

In other cases, the discovery of novel STING inhibitors has been carried out by in vitro screening of libraries of compounds. In 2018, Senlin Li et al. carried out a reporter gene-based screening of a series of cyclopeptides extracted from plants that are well-established ingredients in Chinese medicine and obtained a hit with Astin-C extracted from *Aster tataricus*. Astin-C proved to be able to inhibit STING in vitro, and biotin pull-down assays carried out on the biotinylated compound proved that it bound to the CDN-binding site. Docking calculations were carried out using the structure of human STING of PDB code 4F5D to propose a plausible binding mode in which Astin-C interacted with key amino acids such as L159, S162, and R238 [197]. Recently, a high-throughput screening employing a chip of 3375 compounds (1527 FDA-approved drugs, 795 known inhibitors, and 1053 natural products) was screened against recombinant human STING to identify potential modulators. Quite strikingly, the known Cyclin-Dependent Protein Kinase (CDK) inhibitor Palbociclib was identified as both a human STING and a mouse STING inhibitor. Jiani Gao et al. proved that the reduction of the IFN response was due to STING inhibition rather than the cytotoxic effect of CDK inhibition. They further demonstrated by flagging STING that Palbociclib impaired dimerization, and by additional mutagenesis analysis guided by docking results obtained on the human STING structure found in PDB code 6DNK, that the key residue for Palbociclib binding was Y167 [198].

**Palmitoylation inhibitors and STING-degraders.** Palmitoylation at C88 and C91 is an important post-translational modification of STING as it is responsible for the oligomerization in the lipid rafts in the Trans-Golgi network [44,45]. Interestingly, Nitro-fatty acids (NO_2_-FAs) have been described as bioactive lipids with anti-inflammatory properties and have been found to inhibit STING activation as a response to virus infections (Figure 8 compounds NO_2_-cLA, 9-NO_2_-OA, and 10-NO_2_-OA) [199,200]. In this line, a novel series of small-molecule palmitoylation inhibitors bearing a nitro moiety have been described as STING inhibitors (Figure 8 compounds C-176, C-178, C-170, H-151, BPK-21, and BPK-25) [149,201]. It has been proposed that the nitro moiety conjugated with the double bond increases the rate of the nucleophilic attack of C88 and 91 in the case of NO2-FAs and C91 in the case of the novel small-molecule palmitoylation inhibitors [148].

Making use of the ability of compound C-176 to covalently bind STING, Liu et al. described two CRBN-recruiting PROteolisys TArgetting Chimaeras (PROTACs) designed for the selective proteasomal degradation of STING (Figure 8 compounds SP22 and SP23), which were able to modulate the STING pathway. Of the two designed PROTACs, compound SP23 presented high anti-inflammatory efficacy in acute kidney injury mouse models [202].

It is expected that STING activation mechanisms, especially unconventional activation mechanisms, may lead to the discovery of additional STING ligands or alternative agonists. As many synergistic aspects of STING activation are emerging, the role of the ligands mentioned here, as well as some others, can evolve in the future.

## 8. Perspectives

STING is one of the most interesting targets to modulate the specific activation of type I IFN production. The protein has a well-defined canonical activation mechanism with a characteristic binding pocket to determine specific ligand binding. STING “drugability” may have different agonist applications, including antiviral enhancement, vaccine adjuvant, and antitumor properties, as well as antagonist applications such as repressors of autoimmune activation-related mediators or modulators to reduce trained immunity. Although much information is available about canonical STING activation, refined characterization may determine the specificity in activating some of these properties as compared to others.

The limitations in the use of modified CDNs have prompted the discovery of the drugs mentioned in this text and others not yet disclosed. In addition to proving efficacy, STING drugs require clinical formulations that guarantee the correct delivery and good physiological behavior.

Activation of STING during cell division or events where cellular DNA may be recognized can compromise cell viability. Although DNA sensing by cGAS and IFIT16, among other proteins, can occur at the nuclear DNA level, tight control in the activation of this sensing is important to prevent STING activation [203]. In tumor cells, aberrant control of this process can be associated with an aberrant STING induction, which leads to the induction of type I IFN and the activation of the immune response against the tumor. Tumor immune escape can be determined by the ability of cancer cells to create an immune suppressive environment (i.e., by upregulating programmed death-ligand 1 (PD-L1) expression). Recovering STING activation together with other therapies that overcome immune suppression is a promising antitumor strategy.

Several aspects of STING activation remain still to be characterized. Non-canonical STING activation has been described by different means. The role of ER stress activation, the direct activation through RLRs, the Ca^2+^ signaling, membrane fusion, and conformation, among others, seem to play a role in other STING activation as well. The differences between alternative STING activation compared to CDN activation may represent interesting targets for alternative STING-mediated therapies.

The fascinating journey in unraveling the biology of STING will continue to present surprises in understanding the innate immune response regulation and the tight regulation controlling our response not just to pathogens but also other aspects related to danger signals produced during cellular life, including those associated with ER biology, cell division, mitophagy, or membrane fusion and fission. Along the way, discoveries will lead to applications in human and other animal pathologies.

## Figures and Tables

**Figure 1 ijms-24-09032-f001:**
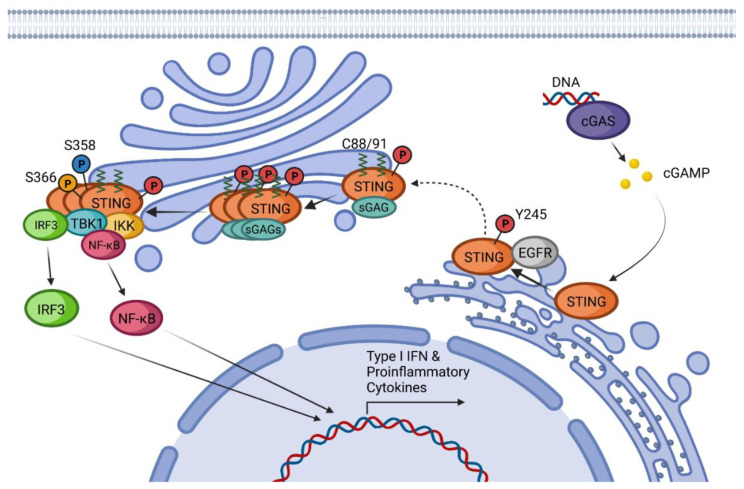
cGAS-STING canonical pathway: Schematic representation of STING canonical activation including post-translational modifications, modulator proteins, and main effectors involved in the production of Type I IFN and proinflammatory cytokines.

**Figure 2 ijms-24-09032-f002:**
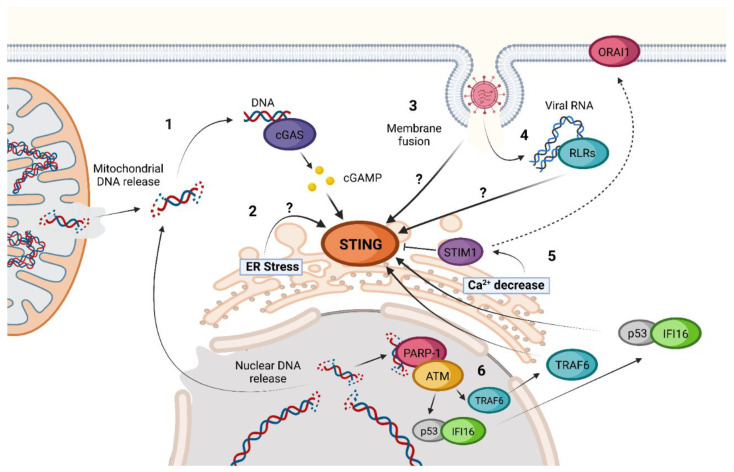
Non-canonical activations of STING. **1.** Release of nuclear or mitochondrial DNA into the cytoplasm during viral infection or after cell damage. **2.** Activation of the STING–TBK1–IRF3 axis by ER-stress inducers such as ethanol. **3.** cGAS-independent activation of STING triggered by membrane fusion. **4.** Activation of STING by RIG-I-like Receptors (RLRs) after detection of viral RNA in a MAVS-independent process. **5.** Ca^2+^ depletion in the ER triggers activation of STIM1 sensor, an inhibitor of STING, and induces its migration to join the calcium channel ORAI1, increasing STING activity. **6.** IFI16 pathway of nuclear-damaged DNA detection in the absence of cGAS activation. Question marks (?) represent “unknown activation mechanism”.

**Figure 3 ijms-24-09032-f003:**
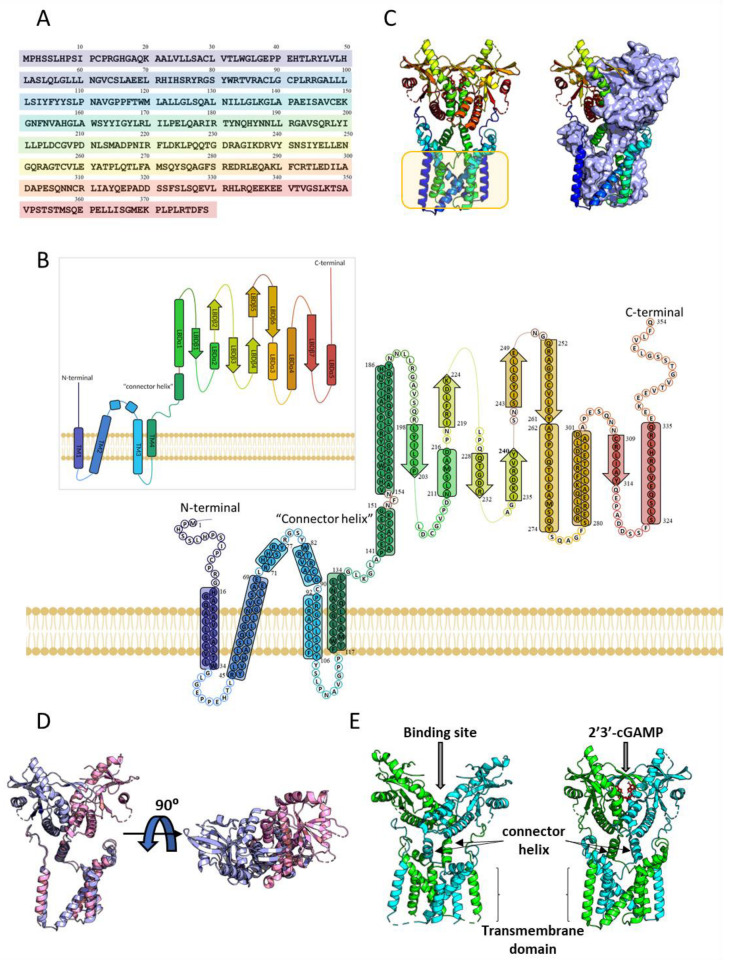
(**A**) Amino acid sequence of human STING. Rainbow color coding indicates the direction from the N-terminal (blue) to the C-terminal (red) ends. (**B**) Schematic representation of the tertiary structure of a human STING monomer with the same rainbow color coding. Amino acid numbering of the secondary structure has been taken from the human STING structure under PDB code 7SII. (**C**) **Left**. PyMOL cartoon representation of human STING as found in PDB code 7SII with rainbow color coding and the 2′3′-cGAMP colored as red sticks. The transmembrane domain is highlighted in yellow. **Right**. PyMOL representation of the closed conformation of the human STING dimer extracted from PDB code 7SII in which one of the monomers is shown as cartoons with the rainbow color coding and the other is shown as a surface to account for the intertwining of the transmembrane α-helixes. For the sake of clarity, the 2′3′-cGAMP is not shown. (**D**) PyMOL representation of the superimposition of two human STING monomers in the open (PDB code 6NT5) and closed (PDB code 7SII) conformations colored in light pink and blue, respectively, depicting the 180° rotation of the cytosolic domain over the transmembrane domain of each of the monomers as seen from the side (**left**) and top (**right**) after rotation of the superimposed monomers 90° in the *Z*-axis. (**E**) PyMOL representation of the open (**left**) and closed (**right**) conformation of human STING as found in PDB codes 6NT7 and 7SII, respectively. In both conformations, each monomer is colored green (chain A) and cyan (chain B). For the sake of clarity, in the closed conformation, only the 2′3′-cGAMP molecule is shown as red sticks, whereas ligand 9IM is not shown in this image.

**Figure 4 ijms-24-09032-f004:**
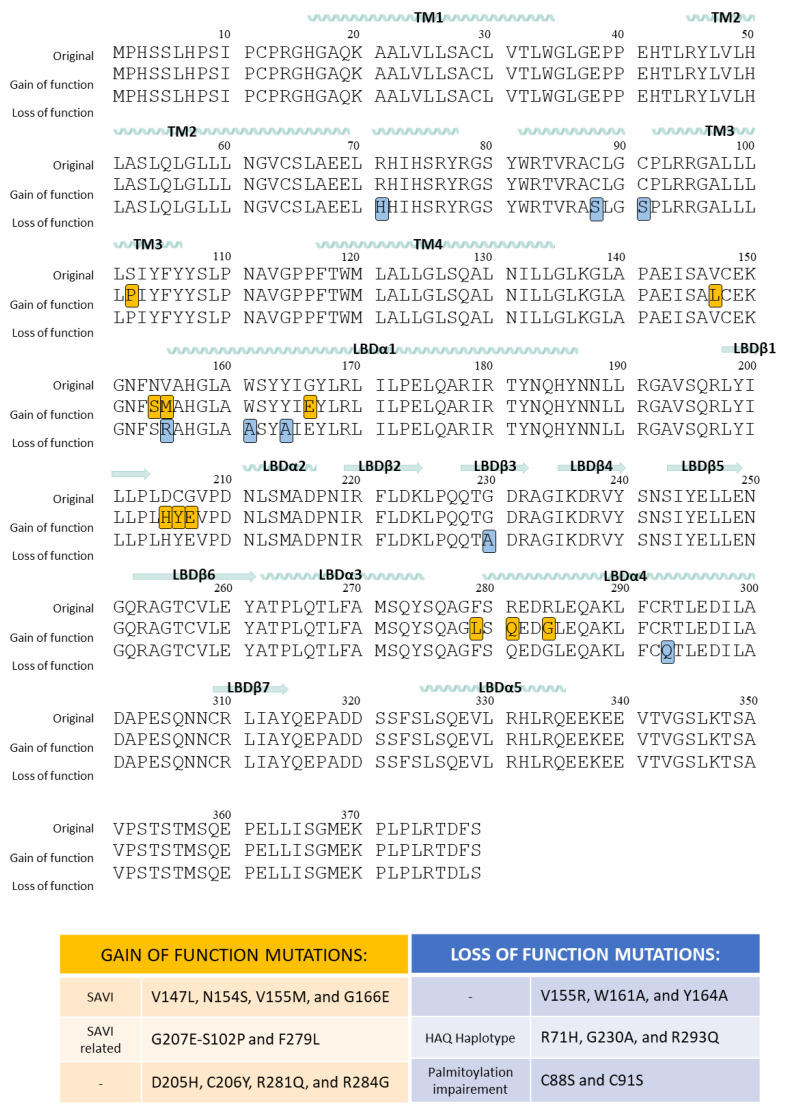
The amino acid sequence of the original human STING and the corresponding mutations that confer gain (yellow) or loss (blue) of the protein function.

**Figure 5 ijms-24-09032-f005:**
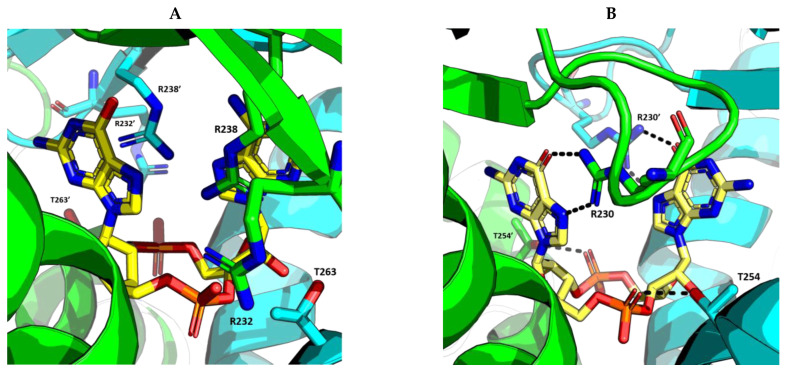
CDN finding to STING: (**A**) PyMOL representation of the binding mode of 2′3′-cGAMP (yellow) to human STING as found in PDB code 7SII. The two side chains of R232 and R238 of both monomers establish ionic interactions with the negatively charged phosphates in the phosphodiester bond. Additionally, and given the position of the side chain of R230 in both monomers, the guanidinium group establishes a π-cation interaction with the nitrogenated base of the 2′3′-cGAMP of the opposite monomer. For the sake of clarity, none of the hydrogen atoms nor the water molecules present in the PDB are shown, and hydrogen bonds are shown as dashed lines. (**B**) PyMOL representation of the binding mode of cdiGMP to the *Myroides* sp. *ZB35* bacterial STING as found in PDB code 7EBL. The side chain of R230 of both monomers establishes very distinct hydrogen bonding interactions that allow the recognition of the guanidine of the cdiGMP, while the side chain of T254 stabilizes the positions of the phosphates in the phosphodiester bond by establishing hydrogen bonds.

**Figure 6 ijms-24-09032-f006:**
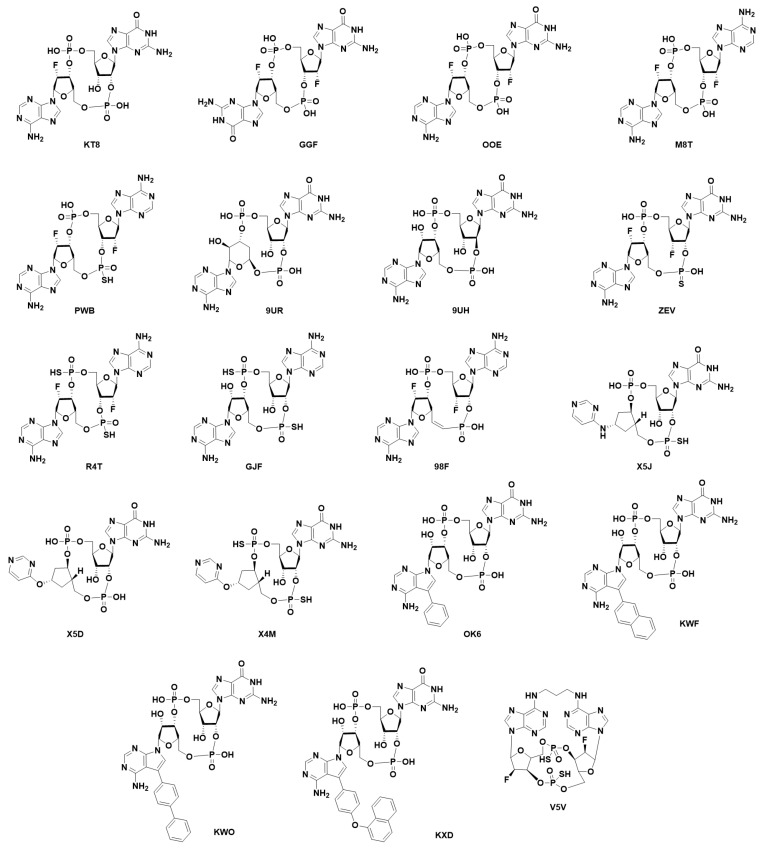
2D structure of the modified CDNs. Compounds that can be found deposited in the PDB in complex with hSTING are identified by the ligand name of the PDB structure and those that are not found crystalized with human STING are identified with the name given in the original article.

**Figure 7 ijms-24-09032-f007:**
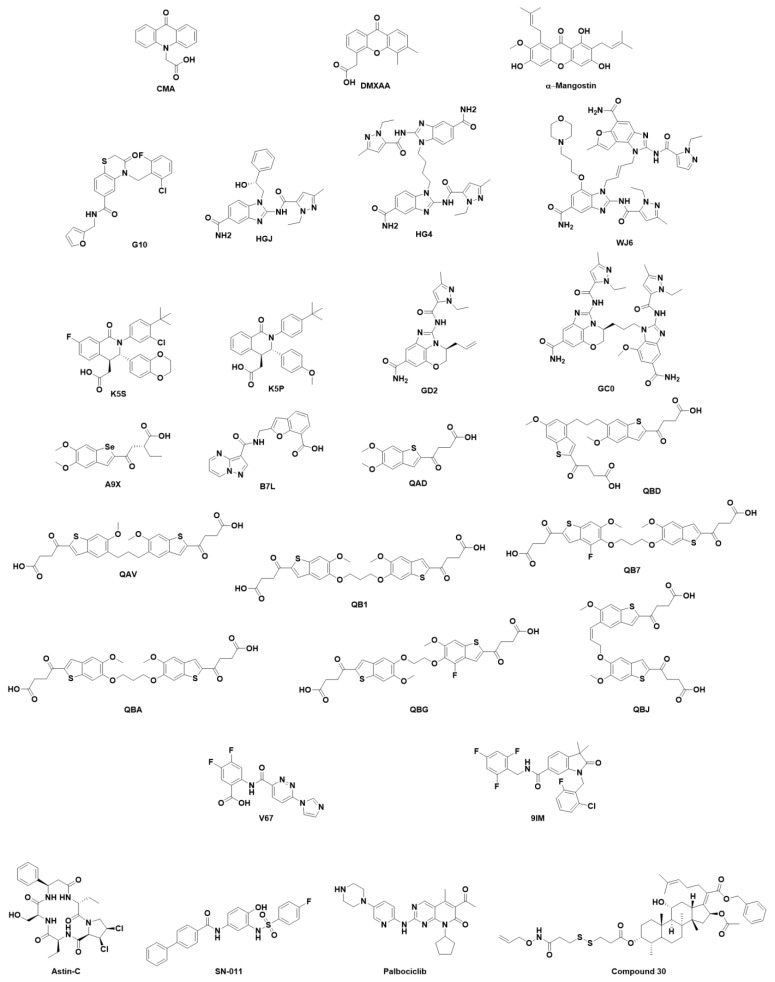
2D structure of the non-CDNs. Compounds that can be found deposited in the PDB in complex with human STING are identified by the ligand name of the PDB structure, those that are not found crystalized with human STING are identified with the name given in the original article.

**Figure 8 ijms-24-09032-f008:**
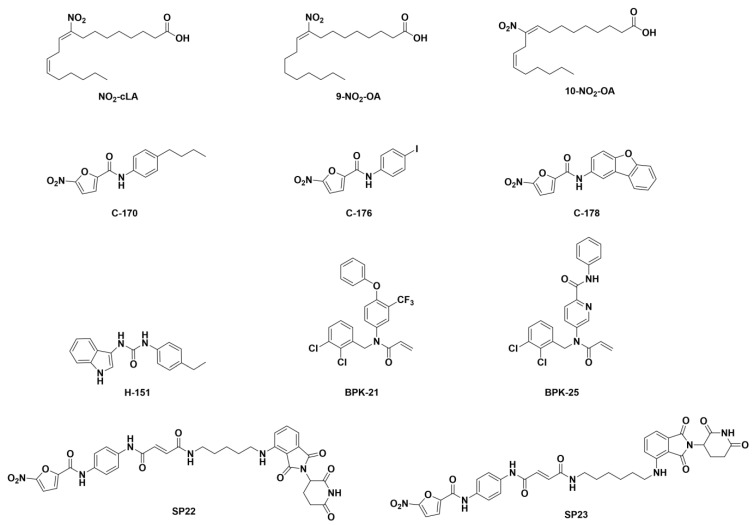
2D structure of the palmitoylation inhibitors and the STING-directed PROTACs. Compounds are identified by the name given in the original article.

**Table 2 ijms-24-09032-t002:** 2D and 3D representations of natural human STING ligands. The 3D structures have been extracted from PDB codes 7SII, 6WT4, 7EBL, and 6IYF (left to right, top to bottom).

2′3′-cGAMP		3′3′-cGAMP	
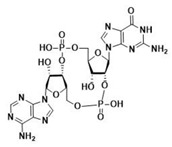	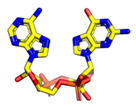	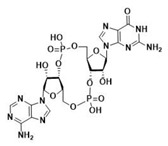	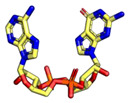
cdiGMP		cdiAMP	
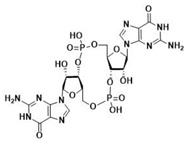	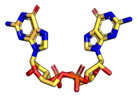	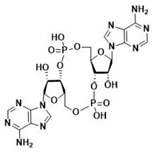	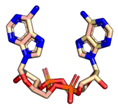

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
