# Peer review of "The Many Ways to Deal with STING"

_ijms, 2023, doi:10.3390/ijms24109032_

Round 1

Reviewer 1 Report

Coderch et al provided a very thorough review on STING canonical pathway, non-canonical pathway, their modulatory mechanisms, polymorphisms, and structural insights.

Overall, the review is comprehensive and well-written. Only point that I suggest is about viral factors. Authors mentioned KSHV but there are a series of publications about EBV and MHV68 (all part of gamma-herpesviruses like KSHV), including ORF52/KicGAS, which seems to be a conserved mechanism to inhibit STING by all gamma herpesviruses. Another point for authors to address is that many of these viral factors are tegument proteins i.e. can be released upon infection and immediately take effects without de novo translation. Authors may want to clarify this point.

I detected minor errors e.g. line#153 "Is well known that" -> "It is well known that..." I recommend through grammar check.

Author Response

Comments and Suggestions for Authors

Coderch et al provided a very thorough review on STING canonical pathway, non-canonical pathway, their modulatory mechanisms, polymorphisms, and structural insights.

Authors’ comments: Thank you for your comments.

Overall, the review is comprehensive and well-written. Only point that I suggest is about viral factors. Authors mentioned KSHV but there are a series of publications about EBV and MHV68 (all part of gamma-herpesviruses like KSHV), including ORF52/KicGAS, which seems to be a conserved mechanism to inhibit STING by all gamma herpesviruses. Another point for authors to address is that many of these viral factors are tegument proteins i.e. can be released upon infection and immediately take effects without de novo translation. Authors may want to clarify this point.

Thank you for your suggestion. As in this review we try to cover STING from different points of view including its activation, regulation, inhibition, and structural bases we have focused only on direct inhibitors of STING activation or its interaction with its known effectors to don’t be too excessive. A sentence stating that only the direct STING inhibitors are included in the manuscript has been added to lines 337-339.

Regarding the different proteins that gamma-herpesviruses encode with the ability to inhibit the cGAS-STING pathway, including ORF52/kicGAS and some proteins forming part of viral tegument, as far as we know, are potent inhibitors of cGAS or act hiding viral DNA from cGAS detection, but have not described as direct inhibitors of STING. For this reason, we have not included them in this manuscript. Nevertheless, as you said, is a very interesting mechanism of inhibition as some of these proteins are released after viral disassembly avoiding the necessity of being translated and would be interesting to cover in future works in which cGAS inhibition is included. Nevertheless, a reference to a review in which inhibition of cGAS is addressed has been added to the text in line 339.

Comments on the Quality of English Language

I detected minor errors e.g. line#153 "Is well known that" -> "It is well known that..." I recommend through grammar check.

Thank you for helping us in the detection of grammar errors. The sentence now at line 157 has been changed to “It is well…” and the main text has been revised.

Reviewer 2 Report

This comprehensive review of STING pathways is much appreciated because it describes canonical and non-canonical STING activations, as well as upregulation and down-regulation of STING pathway.  However, the review needs some modification because there are numerous somewhat incorrect statements that need clarification.

1.       At line 33, the authors refer to pathogenic DNA. Please explain.

2.       At line 36, authors refer to transcription of IFN-beta 1 gene as well as several other genes related to host defense.  This seems to be an understatement. It is more than several other genes. Should state something like “… a multitude of anti-viral gene expressions”

3.       Not sure that STING is the PRR.  cGAS is the PRR and STING is downstream responding to endogenous cyclic diGMP-AMP

4.       At line 145, it is incorrect to infer that type I IFNs expression through IRF3 activation is pro-inflammatory.  IRF3 is involved in expression of ISGs for antiviral innate immune responses and it is involved in expression of several pro-inflammatory cytokines. Please modify statements accordingly near line 145.

5.       Authors make a good point on RNA virus activation of STING. However, the description of the mechanism by which this occurs is vague. Please provide a better description of this important pathway to activation of STING.

6.       At lines 243 to 247, in explaining one aspect of regulation of STING pathway the sentence here is grammatically incorrect and incomplete. Please edit this sentence so that it makes sense.

7.       The authors correctly suggest that several virus target the STING pathway. Please be clear on the known viral factors that directly target STING versus those that target upstream or downstream of the STING pathway. For instance, many viruses target IRF3, which is downstream of the STING pathway. Perhaps a small Table would be beneficial here.

Minor modifications necessary

Author Response

This comprehensive review of STING pathways is much appreciated because it describes canonical and non-canonical STING activations, as well as upregulation and down-regulation of STING pathway.  However, the review needs some modification because there are numerous somewhat incorrect statements that need clarification.

Authors’ comments: Thank you for your comments.

  1. At line 33, the authors refer to pathogenic DNA. Please explain.

Thank you for indicating to us a possible source of misunderstanding. Here we refer to DNA produced by pathogens, including parasites, bacteria, and viruses, as pathogenic DNA. The text in line 33 has been changed to make it clearer “Different stimuli including DNA released by pathogens during infection…”.

  1. At line 36, authors refer to transcription of IFN-beta 1 gene as well as several other genes related to host defense.  This seems to be an understatement. It is more than several other genes. Should state something like “… a multitude of anti-viral gene expressions”.

The sentence at line 36 has been changed to “resulting in the transcription of IFNB1 and a multitude of antiviral and proinflammatory genes.”

  1. Not sure that STING is the PRR.  cGAS is the PRR and STING is downstream responding to endogenous cyclic diGMP-AMP.

We agree with the reviewer that STING only acts as PRR when is involved in the direct detection of bacterial-produced CDNs, in the canonical pathway, STING’s main role is being a platform for signal transduction. In the alternative pathway however, some situations that affect STING direct contact with the stimuli may pose a central role of STING as PRR. We have changed the statement of line 44 to “STING has been described as one of the most important proteins involved in the development of an antiviral response after recognition of pathogen-associated molecular patterns (PAMPs).”

  1. At line 145, it is incorrect to infer that type I IFNs expression through IRF3 activation is pro-inflammatory.  IRF3 is involved in expression of ISGs for antiviral innate immune responses and it is involved in expression of several pro-inflammatory cytokines. Please modify statements accordingly near line 145.

We agree with the reviewer. The sentence has been changed to “TBK1-STING phosphorylated complex recruits and phosphorylates Interferon Regulatory Factor 3 (IRF3) inducing its dimerization and translocation into the nucleus which results in the expression of type I IFNs, a set of Interferon Stimulated Genes (ISGs) and proinflammatory cytokines.”

  1. Authors make a good point on RNA virus activation of STING. However, the description of the mechanism by which this occurs is vague. Please provide a better description of this important pathway to activation of STING.

Thank you for your suggestion. STING triggered under RNA virus infection is recognized in different publications, however, only a few of them provide activation mechanism insights. Trying to solve this issue a better description of the release of mitochondrial DNA under Dengue infection and the nuclear DNA damage under SARS-CoV-2 infection has been added to the manuscript as suggested by the reviewer. On the other hand, molecular mechanisms for fusion and RLR-dependent non-canonical activations of STING remain unknown. A sentence to make it clearer has been added to the main text at line 238.

  1. At lines 243 to 247, in explaining one aspect of regulation of STING pathway the sentence here is grammatically incorrect and incomplete. Please edit this sentence so that it makes sense.

The sentence has been edited to fix the grammar issues and make it understandable.

  1. The authors correctly suggest that several virus target the STING pathway. Please be clear on the known viral factors that directly target STING versus those that target upstream or downstream of the STING pathway. For instance, many viruses target IRF3, which is downstream of the STING pathway. Perhaps a small Table would be beneficial here.

Thank you for your comment. The main scope of the review is covering the existent knowledge about STING from a wide range of fields, including canonical and no canonical activation, cellular regulation, viral inhibition, structural characteristics, and polymorphisms that affect its function. Due to the vast number of viral factors able to inhibit the innate immune response we have focused in the known for acting directly over STING or STING interaction with its effectors. A sentence clarifying this has been added to the main text at line 337.